# Strengthening integration of chronic care in Africa: protocol for the qualitative process evaluation of integrated HIV, diabetes and hypertension care in a cluster randomised controlled trial in Tanzania and Uganda

Marie-Claire Van Hout [iD],[1] Max Bachmann [iD],[2] Jeffrey V Lazarus [iD],[3] Elizabeth Henry Shayo,[4] Dominic Bukenya,[5] Camila A Picchio,[3] Moffat Nyirenda,[5] Sayoki Godfrey Mfinanga,[4] Josephine Birungi,[5] Joseph Okebe,[6] Shabbar Jaffar[6]

**Correspondence to**
Professor Marie-Claire Van Hout;
M.C.VanHout@ljmu.ac.uk

## ABSTRACT

**Introduction** In sub-Saharan Africa, the burden of non-communicable diseases (NCDs), particularly diabetes mellitus (DM) and hypertension, has increased rapidly in recent years, although HIV infection remains a leading cause of death among young-middle-aged adults. Health service coverage for NCDs remains very low in contrast to HIV, despite the increasing prevalence of comorbidity of NCDs with HIV. There is an urgent need to expand healthcare capacity to provide integrated services to address these chronic conditions.

**Methods and analysis** This protocol describes procedures for a qualitative process evaluation of INTE-AFRICA, a cluster randomised trial comparing integrated health service provision for HIV infection, DM and hypertension, to the current stand-alone vertical care. Interviews, focus group discussions and observations of consultations and other care processes in two clinics (in Tanzania, Uganda) will be used to explore the experiences of stakeholders. These stakeholders will include health service users, policy-makers, healthcare providers, community leaders and members, researchers, non-governmental and international organisations. The exploration will be carried out during the implementation of the project, alongside an understanding of the impact of broader structural and contextual factors.

**Ethics and dissemination** Ethical approval was granted by the Liverpool School of Tropical Medicine (UK), the National Institute of Medical Research (Tanzania) and TASO Research Ethics Committee (Uganda) in 2020. The evaluation will provide the opportunity to document the implementation of integration over several timepoints (6, 12 and 18 months) and refine integrated service provision prior to scale up. This synergistic approach to evaluate, understand and respond will support service integration and inform monitoring, policy and practice development efforts to involve and educate communities in Tanzania and Uganda. It will create a model of care and a platform

## Strengths and limitations of this study

► The INTE-AFRICA trial will implement integration of HIV/non-communicable diseases (NCD) services in Tanzania and Uganda in response to an urgent need to respond to increased burden of NCDs, and expand capacity of healthcare systems to manage comorbidity.

► The INTE-AFRICA trial is based on a partnership between African and European researchers, working closely with policy-makers and other stakeholders.

► The process evaluation of INTE-AFRICA employs qualitative and observational methods at two facilities to explore and document stakeholder experiences of integration of services, alongside an understanding of the impact of broader structural and contextual factors.

► The process evaluation works in tandem with quantitative evaluation of clinical efficacy and cost-effectiveness of the INTE-AFRICA trial.

► Limitations of the process evaluation may centre on patient drop-out, characteristics of the two selected sites, selection, information and social desirability bias and other external mitigating factors.

of good practices and lessons learnt for other countries implementing integrated and decentralised community health services.

**Trial registration number** ISRCTN43896688; Pre-results.

## BACKGROUND

Non-communicable diseases (NCDs) have risen rapidly in Africa, alongside a continuing high burden of HIV infection. In sub-Saharan Africa (SSA), HIV/AIDS remains a major

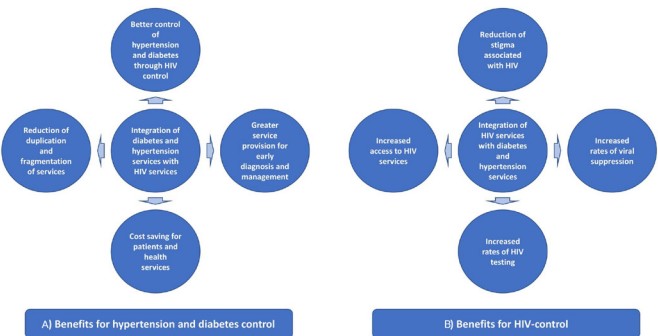

**Figure 1** Potential benefits of integrating diabetes, hypertension and HIV services for (A) DM and hypertension control, and (B) HIV control. DM, diabetes mellitus.

cause of morbidity and mortality among young-middle-aged adults, but the region is also experiencing a rapidly rising burden of NCDs (particularly diabetes mellitus (DM) and hypertension), giving rise to a dual HIV-NCD epidemic.[1–3] While lifestyle changes associated with urbanisation and globalisation (such as eating habits and lack of physical exercise) underpin the ongoing demographic and epidemiological transition towards increasing NCDs, these changes also affect chronic conditions (such as HIV).[1–3] The number of people in regular HIV care is rising,[4] alongside the rate of NCDs in the SSA region, with hypertension representing the single largest risk factor for death, and DM which has seen a massive increase in prevalence in a short period of time.[2 5–7] Patient populations in SSA are increasingly demonstrating younger age of onset of NCDs, with comorbidity of NCDs with HIV, and the impact of NCDs is particularly severe in populations affected by poverty.[3 8–11]

Since 2003, significant global investment and development partner engagement has facilitated the establishment of HIV screening and treatment programmes as the first large-scale chronic disease initiatives in Africa. Health services for HIV are stand-alone and vertically delivered. They have also been combined with decentralisation and task shifting, which has enabled primary health centres to treat large numbers of patients. Non-clinically qualified health workers play a major role in supporting patients on HIV treatment, with almost 70% of people living with HIV-infection in regular care. In contrast, health service coverage for NCDs remains very low.[2 12] Coordinated national NCD control and care programmes are relatively new with significant gaps in funding and operational evidence for programme implementation.[11 13] For example, only about 5%–10% of persons with DM are thought to be in regular diabetes care, and the figure is likely to be similar for persons with hypertension.[2 8 11] Furthermore, those in care experience insufficient health service provision for the diagnosis and management of DM and hypertension (medicines supply is patchy).[2 12 14–17] HIV populations at risk of developing NCDs, and presenting with co/multimorbidities could potentially impact on the gains from the achieved scale-up of HIV care services.[18]

Hence, there is an urgent need to expand the capacity of healthcare systems in the SSA region to provide services for NCDs, either alongside or integrated, with HIV.[19] As chronic conditions, DM, hypertension and HIV-infection require lifelong care. Key challenges to chronic care in SSA are linkage and retention in care, access and medicines adherence.[20 21] As such, the infrastructure and lessons learnt from the HIV chronic disease model can serve as important resources for the expansion of NCD prevention, care and treatment. HIV chronic care management pathways, resources and infrastructure can be leveraged to integrate with newly developing NCD services, ultimately to strengthen the platform for NCD services and improve health outcomes for people with NCDs.[22–24] Health systems have developed the experience managing HIV as a chronic disease, including linking and retaining patients in care and supporting treatment adherence in drugs, diagnostics procurement and other key health systems indicators.[21 25–27] Integration can reduce duplication and fragmentation, streamline services by treat those with co/multimorbidities and potentially offer patient benefits relating to time and cost (figure 1). It could, however, also threaten service capacity by resulting in increased service demand and loss of clinical focus on one disease.

Despite the increase in academic and clinical interest in HIV/NCD integration in SSA,[9 26 28–33] little evidence about integration in terms of both scope and generalisability exists. Extant evidence is limited to small scale feasibility studies in largely different contexts,[34–40] despite suggesting that integrated care is an efficient use of resources compared with the standard of care and beneficial for patients' NCD and HIV clinical outcomes.[28 30 32] There is a lack of large scale or randomised and/or controlled evaluations and context-specific clinical, cost-effectiveness and process outcomes data constrains policy-making and development of integrated care models, which could strengthen health systems when tailored to the distinct needs of each specific SSA country.[19 21 37] Such evidence is paramount to inform policy, government resource prioritisation and to develop integrated care models which strengthen health systems best suited to the needs of that specific SSA country. The INTE-AFRICA trial is conducted to respond to the insufficient existing evidence to substantiate the benefits and sustainability of integrated care as well as the implementation of such programmes in the SSA setting. We present here the qualitative process evaluation protocol designed to evaluate INTE-AFRICA, a European Commission Horizon 2020 funded implementation research project (protocol number 19–100) operating in Tanzania and Uganda.

## The INTE-AFRICA trial
INTE-AFRICA aims to implement and assess the effectiveness of the integration of HIV, DM and hypertension services at the point of service delivery covering many health facilities where common approaches to clinical decision making, drug procurement and human resource

| Table 1 | key data on the country settings | |
| --- | --- | --- |
| | **Tanzania** | **Uganda** |
| Income level | Low | Low |
| Population size | 58 m(2018) | 35 m(2016) |
| Estimated prevalence of hypertension from STEPS survey | 26% | 26% |
| Estimated prevalence of diabetes from STEPS survey[15]* | 5%–10% | 2%–5% |
| Estimated prevalence of HIV-infection | 5.1% (2017) | 6.2% (2017) |
| Doctors density/100 000 population | 3 (2014) | 0.8 (2005) |

*Diabetes estimate varies according to age and gender. Data are of variable quality but reference 11 shows that the overall median diabetes prevalence in 12 countries in Africa is 5%.

management occur. The project will generate the research evidence needed by health services in Africa, to scale up and sustain chronic disease management and services in an integrated manner. We are also interested in observing changes in dynamics pertaining to stigma and discrimination associated with HIV.[41]

INTE-AFRICA team specifically chose Tanzania and Uganda as they are low-income countries and their public and private health facilities are strongly committed to providing services for NCD. However, their health systems struggle to scale up provision for diabetes and hypertension in the face of competing health demands, including HIV-infection (table 1). Tanzania and Uganda also share relevant characteristics with other countries in SSA. As such, the process evaluation has the potential to enhance the generalisability of integrated care to other similar settings by providing understanding of the determinants and mechanisms of the implementation process.

Our programme is underpinned by a participatory, multiactor approach which supports dialogue and knowledge exchange, fosters mutual understanding and provides input in policy agendas around diagnosis and treatment of DM, hypertension and HIV in an integrated clinic. The INTE-AFRICA conceptual framework is illustrated in figure 2.

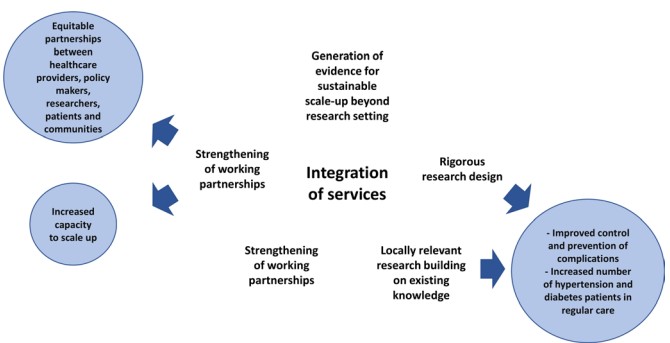

**Figure 2** INTE-AFRICA conceptual model.

INTE-AFRICA will test integrated health services at primary care centres for HIV-infection, diabetes and hypertension, by providing a 'one-stop' integrated care clinic for these conditions (the intervention). We will conduct a pragmatic parallel arm cluster randomised trial: 32 largely urban health facilities offering primary care services in the two countries will be randomised, with 16 facilities allocated to deliver the intervention immediately (intervention arm), and 16 facilities to continue with usual care (control arm). At each selected facility, cohorts of approximately 220 patients; 110 HIV infected and 110 NCD patients, will be enrolled to evaluate the primary research outcomes.

The primary care facilities will initiate and stabilise patients on treatment, manage complications (including referral to higher facilities) and conduct clinical and laboratory monitoring of patients for all three conditions. Specific characteristics of integrated care in each 'one stop' integrated care clinic are: concurrent management of HIV, hypertension and DM in the same facility; management of patients with HIV, hypertension and DM by the same clinician or team of clinicians (nurses, counsellors other staff); integrated training of clinicians; single waiting area and queue; integrated health education about all three conditions; one pharmacy with a single drug dispensing point; similar testing and cross-testing, for diagnosis and monitoring with the requisition of laboratory tests in the same place; and similar format of paper medical records for each condition kept in the same patient folder.

Patients who decline to participate in the research and trial participants in control arm clinics will continue to receive standard vertical healthcare delivery. Inclusion criteria for participation are: being over 18 years old; having confirmed HIV infection, DM or hypertension; living within the catchment population of the health facility; likely to remain in the catchment population for 6 months and willing to provide written informed consent. Very ill patients requiring in-patient care will be excluded. The research team will observe participants during clinic visits with additional reviews at 6 and 12 months. Further details are provided in the full INTE-AFRICA trial protocol.

### Qualitative process evaluation of the INTE AFRICA trial
The aim of the process evaluation of INTE-AFRICA is to explore the experiences, attitudes and practices of a wide variety of stakeholders during the process of programme implementation and to develop an understanding of the impact of broader structural and contextual factors on the implementation process of service integration. Process evaluations typically evaluate how and whether interventions are delivered as intended and whether such implementation is congruent with the theory underpinning the intervention.[25 42–44] Updated Medical Research Council guidance for evaluation of complex health interventions has recently recognised the value of process evaluation within trials stating, 'it can be used to assess fidelity

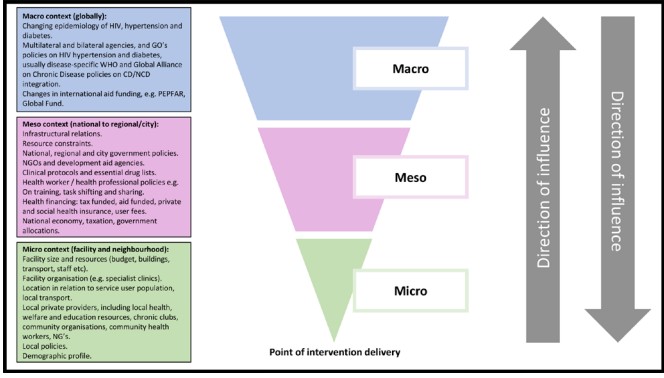

**Figure 3** Potential contextual influences on INTE-AFRICA programme implementation cascade. CD, chronic disease; NCD, non-communicable diseases; NGO, non-governmental organisation; PEPFAR, President's Emergency Plan for AIDS Relief.

and quality of implementation, clarify causal mechanisms and identify contextual factors associated with variation in outcomes.'[25] Hence, the process evaluation in INTE-AFRICA is particularly focused on context, description of the intervention and its causal assumptions, implementation, mechanisms of impact and outcomes.[25] Further we evaluate the extent to which resources and activities supporting the intervention function to deliver intended outputs, with subsequent improvements in outcomes.

A central focus lies in identifying contextually relevant strategies for successful implementation of service integration, and practical difficulties in adoption, delivery and maintenance to inform wider implementation.[45] We recognise that outcomes (eg, knowledge gained) of INTE-AFRICA are dependent on understanding cultures and contexts of the stakeholders (ie, patients, healthcare providers, policy-makers, community leaders/members, non-governmental organisations (NGO), international organisations and clinical researchers) involved and those surrounding service design and delivery as well as care seeking practices.

The role of social and behavioural science approaches to understanding individual experience, and the role of such contextual dynamics is central to this process evaluation. It aims to enhance understanding of issues related to the NCD and HIV agendas and service delivery approaches from all perspectives and stakeholders along the integration process. INTE-AFRICA will use a broad social behavioural approach to support community engagement in research in Tanzania and Uganda, going beyond clinical trial recruitment and retention to improve NCD/HIV literacy. Behavioural and social science research strategies[46] will allow NCD and HIV-related research to place the needs and perspectives of people living with NCD and/or HIV at the centre of the progression of clinical studies, such as INTE-AFRICA, and investigate participant motivations and decision-making processes related to preferences for or participation in different types of integrated and decentralised services.

### Theoretical framework: Bronfenbrenner's ecological model of behaviour

Process evaluation design, like healthcare interventions, requires a theoretical framework to structure the qualitative and observational evaluation across sites. The chosen theoretical framework is Bronfenbrenner's ecological model of behaviour[47 48] used to conceptualise integrated care as events that disrupt complex social systems[49] operating across multiple contextual levels. We are interested in better understanding the importance of context, which becomes especially relevant when comparing Tanzania and Uganda. For example, healthcare coverage and charging dynamics between countries and between HIV as opposed to NCD care differs. In Tanzania, while HIV services and drugs for HIV are free, patients are required to pay a direct user fee (unless elderly or very poor) or use their health insurance to pay for NCD drugs. Currently, there is a strategy to progress to single health insurance to support universal health coverage in Tanzania. In Uganda, HIV services are also free to all (public and private not for profit health facilities) and HIV drugs are always in stock. However, in Uganda while NCD services are free in public facilities, stockouts of laboratory reagents and NCD drugs are frequent, and patients have to pay a direct user fee to access them. There is also no national insurance scheme in Uganda, only private insurance which is relatively expensive and optional.

Using the Bronfenbrenner theoretical framework, contextual factors are evaluated at the macro (universal vs partial coverage; free primary healthcare and drugs, affordability and users fees; fiscal policy, financial aspects at government level, funding barriers for chronic care management and barriers to government scale-up of service integration); meso (HIV/NCD drug ordering, drug delivery systems and continuity of supply, healthcare provider education and employment, community understanding and perspectives on multimorbidity) and micro (clinic and pharmacy level management, resources, patient/service user experience) levels (figure 3). These contextual factors, likely to influence implementation of integration and their effects, will be investigated to capture variation in adoption, delivery and maintenance outcomes as well as responses to the intervention. This will affect both reach and fidelity, which are hypothesised to be important factors in outcome differences.

To situate INTE-AFRICA within the Bronfenbrenner theoretical framework, we will develop a logic model. This model will set out contextual determinants of HIV and NCD care management in SSA and assess how integrated care components function to address these determinants to improve outcomes in the management and support of patients (table 2).

### METHODS
#### Study setting
This will be a cohort study taking place in one 'one-stop' integrated care clinic per country. In Tanzania, the

**Table 2** Logic model of programme inputs, processes and outcomes

| Intervention inputs | Changes in care processes | Outcomes |
|---|---|---|
| Negotiation with national, district and local government health departments, NGOs and funders | Agreement about and support for service model, including reorganisation of clinics and staff<br>Commitment to ensure drug supply | ► Reorganisation of clinics and staff to implement the model<br>► An effective, quality and sustainable funded drug supply chain |
| Negotiating lower drug prices<br>Supporting and monitoring drug ordering in each clinic<br>Providing buffer drug supplies to each clinic | Drugs always in stock | ► Increased diagnosis of comorbid conditions<br>► Increased retention and adherence<br>► Increased viral suppression, better control of blood pressure and blood glucose<br>► Less AIDS, cardiovascular disease and diabetes complications<br>► Lower patients costs (travel and absence from work)<br>► Less health service duplication and costs (health service costs might increase if more patients are diagnosed and are more adherent)<br>► Increase patient satisfaction<br>► Increased clinician satisfaction; less burn-out and absenteeism<br>► Reduce missed opportunities for improving care and health outcomes |
| Engagement with and support for clinicians and managers in each clinic | Clinicians and managers enable and support integration and find solutions to emerging problems | |
| Provision of integrated service in each clinic (alongside and additional to existing services) | Trial participants attend integrated service<br>Avoid multiple visits for patients with multimorbidity | |
| Training clinicians about integrated clinical management | Better diagnosis and treatment including attention to comorbid conditions | |
| Community engagement | Identify and enlist community organisations and resources to help with health education, tracing defaulters or patients who have difficulty attending clinic | |
| Providing standardised stationery for integrated medical records; training clinicians to use it | Increased awareness by clinicians and patients about disease severity, comorbidity, adherence and control in individual patients | |
| Improving monitoring and evaluation based on clinics registers and medical records | Regular data analysis and feedback to staff | Quality assurance and continuous improvement in the quality of care |
| Identifying effective health education | Improve health education at clinics | Healthier lifestyles<br>Increased adherence |

NGO, non-governmental organisation.

selected site is Temeke Regional Referral Hospital in Dar Es Salaam, a public tertiary health facility, with 465 staff, and serving a population of over 1 million. In Uganda, the selected site is the Kasangati Health Centre IV, Kasangati, a public district health facility located in the Wakiso District serving a population of over 2 million. Both provide a range of services to the community (primary care, HIV/AIDs, NCDs, surgical, maternal and child health, health education, dental and pharmacy).

## Study design
This protocol describes procedures for a qualitative process evaluation of the INTE-AFRICA pragmatic parallel arm cluster randomised trial comparing integrated health service provision for HIV-infection, DM and hypertension with the current standard vertical care delivery model. The process evaluation works in tandem with the collection of selected clinical outcomes (eg, clinical efficacy of different treatments) and health economic data (eg, costs and benefits of different approaches) to estimate the potential benefits to patients and health services at clinic and country level (protocol reported elsewhere).

## Patient and public involvement
Patient and public involvement (PPI) throughout a programme of research enhances research quality and relevance by providing different perspectives and a sense of ownership. This protocol will adhere to the same principles, and will allow the voice of 'service users' and those affected to be heard, and utilised. Key stakeholders such as patients as service users and their families will be fully involved in guiding the research, acting as research participants, and in implementation of change in health service delivery and integrated care planning. All aspects of the process evaluation are underpinned by participatory action health research and its success and usefulness will be grounded in PPI, participation and engagement in the form of patient/professional identification of research priorities, collaborations and partnerships, expert steering, community participation around health needs and optimal integrated services, awareness raising activities, development of print materials, toolkits and training for healthcare professionals.

It will use three qualitative research techniques: in-depth interviews with stakeholders (patients, healthcare

provider, policy-maker, NGO/international organisation, and clinical researcher); focus group discussions (FGD) with community leaders and community members; and clinic-based observations in one site per country. This design permits an assessment of the fidelity of INTE-AFRICA's implementation, a detailed description of the processes, relationships, and contexts involved in the delivery of integrated care, and the identification of factors attributing to the failure or success of the programme. It thus addresses the 'black box' problem in interpreting trial results by improving understanding of the mechanisms that connect particular intervention components to particular outcomes.[50] The chosen approach will enhance social construction and acceptability of chosen decentralised integrated approaches, link outcomes to policy and advocacy and impact sustainability of HIV, DM and hypertension chronic disease service integration in two low and middle income countries. It provides the opportunity to document and refine INTE-AFRICA activities prior to a larger pragmatic trial or scale-up by Tanzanian and Ugandan governments. These synergistic approaches to evaluate, understand and respond will support integration, support affordability, address barriers to government scale up and funding barriers for chronic care, inform surveillance, policy and practice development and improve efforts to involve and educate communities in Tanzania and Uganda. It will create a model and a platform of best practices and lessons learnt for other countries implementing integrated and decentralised community health services for HIV and chronic disease. The process evaluation methods for each objective are described in table 3, including how each method maps onto the three different contextual levels.

### Study population and recruitment

wenty-five patients and 10 healthcare providers in Temeke Regional Referral Hospital in Dar Es Salaam, Tanzania and Kasangati Health Centre IV, Kasangati, Uganda will be purposely selected and invited; at 6, 12 and 18 months after the start of the trial, to reflect retrospectively on their experience of integration. Healthcare providers include the hospital/health centre overall in charge and pharmacist, and the 'one-stop' medical officers in-charge, trained clinicians managing HIV, diabetes and hypertension patients, pharmacist, laboratory technician, counsellors or nurses providing health education

**Table 3** Process evaluation design and data collection framework

| Post-Integration. Data Collection at each Site* | 6 months | 12 months | 18 months | Contextual level |
|---|---|---|---|---|
| Observations of consultations, different processes and clinic flow at clinic levels and in non-clinical areas. | 1 week | 1 week | 1 week | Microcontext (facility and neighbourhoods) |
| In-depth phenomenological interviews with patients/ service users | 25 | 25 | 25 | Micro context (facility and neighbourhoods) |
| In-depth phenomenological interviews with healthcare providers at the clinic (hospital overall in charge, hospital pharmacist, the medical officers in-charge of the integrated clinic, trained clinicians managing HIV, diabetes and hypertension patients, pharmacist, laboratory technician, counsellors or nurses providing health education and counselling and nurses in the registration desk who are also responsible in taking vital signs) | 10 | 10 | 10 | Mesocontext (national to regional/city) Micro context (facility and neighbourhoods) |
| Semistructured interviews with Ministerial policy-makers and provincial/regional/district level clinical/health senior management (Director for NCD, HIV and curative services). | – | 5 | 5 | Macrocontext (Global) Meso context (national to regional/city) |
| Semistructured interviews with NGO and international organisations (eg, WHO Country office, UNAIDS, PEPFAR, CDC) | – | 5 | 5 | Macrocontext (Global) Meso context (national to regional/city) |
| Focus group discussions (FGD) with community leaders (8–12 participants) | 1 | 1 | 1 | Microcontext (facility and neighbourhoods) |
| FGD gender specific with community members (8–12participants) | 2 | 2 | 2 | Microcontext (facility and neighbourhoods) |
| In-depth phenomenological interviews with clinical researchers | | | 4 | Microcontext (facility and neighbourhoods) Meso context (national to regional/city) |

*The * refers to 'Numbers indicated per country'.
CDC, Centers for Disease Control and Prevention; NGO, non-governmental organisation; PEPFAR, President's Emergency Plan for AIDS Relief; UNAIDS, Joint United Nations Programme on HIV/AIDS.

and counselling, and nurses in the registration desk who are also responsible in taking vital signs from patients. We will also collect qualitative data from interviews with Ministerial policy-makers and provincial/regional/district level clinical/health senior management (directors for NCD, HIV and curative services); NGO and international organisations (eg, WHO Country office, Joint United Nations Programme on HIV/AIDS (UNAIDS), President's Emergency Plan for AIDS Relief (PEPFAR), Centers for Disease Control and Prevention (CDC)) and clinical researchers; and conduct FGD with community leaders and community members. These numbers are expected to reach saturation (ie, the point that further information does not provide any additional variation in observed themes). We may replace participants, for example patients if there is significant loss to follow-up or refusal for repeated interviews. For instance, if a participant drops out at 12 months, we still have their 6-month experience documented, and we can replace with a new participant, invited to reflect on their 12-month retrospective experience. Where possible we will gender match interviewers with participants (particularly patients). Participants (patients, healthcare providers and others) will not be directly compensated bur rather they will be compensated for incurred transport costs to attend the interview/FGD, and provided with refreshments during the interview/FGD.

The following recruitment procedures will take place at each 'one-stop' integrated care clinic:

► Observations will be made in the integrated clinic in consultation with the hospital/health centre and 'one-stop' clinic in-charges.
► Recruitment of patients will be supported by clinic nurses who will identify and approach selected participants who have a minimum of 6 months experience of integration, and will be asked to consent to partake in an in-depth interview on the day they attend the clinic. INTE-AFRICA researchers will purposively sample women and men of different ages and explore any age/gender and condition ((HIV/hypertensive/DM/multi/comorbid) related differences.
► Healthcare providers at the integrated clinic will be approached to participate in an in-depth interview on the day the INTE-AFRICA team are scheduled to attend.
► Ministerial policy-makers and provincial/regional/district level clinical/health senior management will be identified and requested to participate in a semistructured interview (face to face, online using Zoom or telephone).
► NGO and international organisations will be identified and requested to participate in a semistructured interview (face to face or telephone).
► Community leaders will be identified in the clinic catchment areas by virtue of their position while community members will be identified in consultation with the community leaders and invited to participate in the FGDs.

► Clinical researchers will be invited to participate in an in depth interview at the 24-month end point.

## Data collection

We will use the Empirical Phenomenological Psychological (EPP) five-step method,[51] which combines psychological, interpretative and idiographic components, to collect data. The data will garner an understanding of the complex social processes, of social, aged, gendered and culturally/community specific meanings and broaden the incremental understanding of the distinct lived experience of policy-makers, patients, healthcare providers, researchers and communities. We will balance the description of phenomena with the interpretation of insights and are cognisant of participant experiential phenomena and authors' interpretation of associated meanings. It will yield an in-depth sociocultural understanding of patient-reported/participant-reported outcomes, their motivations, preferences, beliefs, expectations, identities, hopes and views on conditions, related stigma and of decision-making processes. This will provide better understanding of stakeholder and community positioning during integration. This understanding will inform policy and practice, ensure effective patient/service user education, position service users and their families to understand these conditions and interpret study outcomes and facilitate future HIV and chronic disease clinical studies.

Both descriptive patient-level data and rich sociobehavioural qualitative/observational data will be collected by a team of trained researchers in Tanzania and Uganda. Data collection will entail exploring the experiences, attitudes and practices of a wide variety of stakeholders during the process of INTE-AFRICA programme implementation and will develop an understanding of the impact of broader structural and contextual factors on the implementation process.[10 45] We will collect data on social behavioural and cultural aspects impacting implementation (eg, individual and community health risks, protective behaviours and health responses) within the broader social and political frameworks (government resources and barriers to sustaining integration), the practicalities of accessing, providing and sustaining integrated services (eg, staff time, resources, equity of access, supply chain dynamics and pharmacy components pertaining to drug types, drug/reagent availability and costs for integrated patients, catchment area/populations, quality of care, waiting room dynamics, record keeping and retention across multi/comorbidities, training gaps); and process indicators (eg, perceived stigma, acceptability of vertical vs integrated service designs, lay knowledge and awareness, the dynamics of public vs private sector integration (where relevant to the participant), and bottlenecks to accessing services). We will also describe implementation of the intervention in terms of fidelity to the intended model of care, adaptations to the intervention during implementation, and dose and reach of intervention components actually delivered and received (such as numbers and proportions of eligible staff who received

integrated care training, numbers of proportions of patient participants who received all or most of their care from integrated services, and frequency of drug stockouts). The latter data will be complemented by routinely collected quantitative data such as training attendance and medical records. We will document changes in healthcare provider roles, attitudes and patient relationships.

Interviews with patients/service users, healthcare provider and policy-maker/senior manager will include specific questions about their experience and management of individuals with multimorbid HIV, hypertension and/or diabetes. These include their perceptions of INTE-AFRICA; impacts of INTE-AFRICA on the provision of integrated HIV/AIDS care and NCD care, and relationships with NGO and international organisations; changes in health provider roles, attitudes and patient relationships; impacts of the INTE-AFRICA implementation context on trial and health economic cost outcomes; impacts of the INTE-AFRICA intervention on an integrated health systems approach to care (medicine supplies, record keeping, service user education, clinical care pathways, data management, staff training); and barriers to and facilitators of change and future sustainability of integrated care provision. We will assume a more pragmatic approach when garnering perspectives from higher level stakeholders involved in health policy and practice generation, and NGO and international organisations (eg, WHO country offices, UNAIDS, PEPFAR, CDC) providing peripheral supports and guidance. We are especially interested in better understanding the complexities around government scale up and resource allocation for chronic care (eg, decentralisation, financial planning, identification of potential funding sources at ministry levels, subsidised NCD drugs and by international donors (CDC, PEPFAR, UNAIDS) (table 3).

## Data analysis and synthesis

The analysis of qualitative data will be iterative, moving between data collection and analysis to test emerging theories. Field notes of observations will be analysed thematically to provide a description of the process and content involved in adapting and delivering the intervention. Audio recordings of interviews and FGDs will be transcribed verbatim by competent and experienced social scientists, with a subsample transcribed using conversation analytic conventions. Translation from local languages (eg, Swahili, Luganda) into English will be performed for easy sharing with the study partners. Translation will occur using a back-translation method for consistency. An electronic data management package (eg, NVivo) will be used to manage the qualitative data analysis at the respective country levels. The analysis of the observational data will require knowledge obtained from health professional interviews at different levels to compare how reported experience, and different accounts of patient and professional perspectives relate to actual implementation of INTE-AFRICA scenarios: (1) when DM and hypertension services are integrated with

HIV-infection services and (2) comparing countries. Care will be taken to identify and follow up deviant cases which do not fit into emerging theories. Reliability and validity of the analysis is optimised through iterative data collection, the use of a multi-method design incorporating interviews, FGD and observations and the ongoing discussion of findings within the research team for scrutiny and feedback.[52 53]

The chosen phenomenological approach (EPP) to collecting and analysing data, usually used in psychological research, reveals the structures of subjective experience and meaning of a lived phenomenon (in first person point of view). It follows to some extent Husserl's principle of active efforts to 'bracket out' the researchers' theoretical preunderstanding in the first steps of a text analysis.[51] The 'bracketing', however, does not exclude an empathetic, psychological focus in the analysis on the experiences of the researched phenomenon as it is lived by the informant and what it means is to her or him. In the context of INTE-AFRICA, researchers in both countries will strive to get an empathetic understanding of the text, and hence do not apply their professional prior knowledge about integration. The analysis of the observational data will also require knowledge from health professional interviews to compare how reported experience relates to actual implementation of integration at the clinic level.

We will conduct a stepwise EPP analysis in five steps. First, the text will be read several times to get a good grasp of how the informant spoke about the researched phenomenon of integration. In this step, theoretical reflection will be withheld. Second, the whole text will be divided into meaning units of a whole paragraph or a single word. Third, the informant's personal language will be transformed, unit by unit, to the researchers' language. The researchers will discuss the transcription unit by unit. When different interpretations occur, the researchers will return to the interview text and discuss in a free, imaginative process until agreement can be reached through negotiated consensus. Fourth, the text will be screened in a search for comprehensive themes. The text will be interpreted with connection to the researchers' theoretical knowledge in an interchange between the original data, the transformed units and the researchers' theoretical preunderstanding about integration. The meaning units will be assorted into appropriate themes and thus constitute a general structure of the phenomenon of integrated care. Fifth, this essential structure will penetrate all the revealed themes and thus the meaning of the researched phenomenon of integrated care to the informant.

## Credibility and transferability

The process evaluation protocol adheres to recommendations intended to facilitate the standardisation of process evaluation design and reporting.[43] It provides a unique opportunity to document implementation and collaboratively refine integrated care in two SSA countries. This makes possible the synthesis of results of similar studies elsewhere in the SSA region in future. In order to ensure

credibility, while we use different methods of data collection (qualitative/observational), and operate concurrently with clinical outcomes data and health economics analysis, we will also add a further layer of triangulation of sources in terms of perspectives across stakeholders and across conditions (HIV/hypertension/DM/multi/comorbidity) when raising the abstraction level. Triangulation of sociobehavioural qualitative and observational data during analysis will occur in order to understand how different types of evidence enhance the overall interpretation of how INTE-AFRICA was implemented, and what the additional health economic and clinical data are, drawing case comparisons across clinics and across countries, and developing possible explanations for implementation variation. The data, when combined and triangulated across these multi stakeholder perspectives, will provide a 'thick description', of how the intervention was delivered, maintained and experienced by stakeholders.[10 45] It will also offer explanations for observed variation over time and between countries, and detailed insight into the interaction between different contextual features and components of integration of NCD or HIV/NCD services. It will also facilitate triangulation of information across stakeholders, clinics and countries. This approach will help to support transferability to other settings, by identifying factors which are plausibly and/or consistently related to successful or unsuccessful delivery of intervention components. We recognise the potential for selection and information bias as limitations of the trial itself, and mitigate by using a random sampling approach, defining characteristics in a cohort, using a standardised approach to collecting data with continual assessment of information bias, and ensuring that research personnel are unaware of participant disease status. We will address social desirability in the process evaluation by only providing brief information at the outset of the evaluation in order to avoid priming, using an interview schedule approved by a panel of INTE-AFRICA experts in terms of sensitivity, conducting qualitative research using skilled interviewers with limited power relationship between interviewer and participant, conducting the interviews in a safe and secure setting where the participant feels comfortable, briefing them that there is no right and wrong answer, and finally by encouraging them to use anecdotes and experiential evidence to support their views.

Emerging theories and the relationship of the data to the conceptual literature underpinning the intervention will be discussed and refined at INTE-AFRICA research team meetings throughout the project. We envisage utilising the public understanding of science theory[54] to unpack how patients and their communities in Tanzania and Uganda understand and use different knowledge on HIV and NCDs in their lives. This could be facilitated by understanding how they create meaning from scientific findings relating to NCDs and HIV and if, how, and to what degree they incorporate these findings into their everyday lives. This theory has the capacity to shift public attitudes by connecting and communicating the development of innovative scientific concepts in the medical field (in this instance integration of HIV/NCD services in Tanzania and Uganda) to the non-scientific public, and thereby enhance education, training cascade, health policy and practice, and ultimately public understanding of multimorbidities and sustainable routes to care. Further, it will create a platform for the sharing of lessons learnt, best practices and context adaptation of the final integrated model of care in other African countries (clinical care policies and practice, staff cascade of training, service user education and community awareness raising).

## ETHICAL CONSIDERATIONS

Ethical approval for the evaluation has been granted by the research ethics committees of the Liverpool School of Tropical Medicine (UK), the National Institute of Medical Research (Tanzania) and TASO Research Ethics Committee (Uganda). The key ethical principles of voluntary and informed participation, confidentiality and safety of participants will be used in all researcher and participant interactions. Written consent for interviews and observations will be obtained from all participants. All participants will be provided with written information about the research, this will be explained verbally, and informed that their participation is voluntary and that they may withdraw from participation at any time. Safety and confidentiality of all data will be ensured by: (1) encrypting all transcriptions with a password-protected code; (2) storing all data in a secure, encrypted database accessible only to authorised persons on the research team; (3) delinking all personal information of participants from the data collected and stored. Each participant will have a unique identification number.

### Author affiliations
[1]Public Health Institute, Liverpool John Moores University, Liverpool, UK
[2]Norwich Medical School, University of East Anglia Faculty of Medicine and Health Sciences, Norwich, UK
[3]Barcelona Institute for Global Health (ISGlobal), Hospital Clinic, University of Barcelona, Barcelona, Catalunya, Spain
[4]Muhimbili Centre, National Institute for Medical Research, Dar es Salaam, Dar es Salaam, Tanzania, United Republic of
[5]MRC/UVRI/LSHTM Uganda Research Unit, Medical Research Council Uganda, Entebbe, Uganda
[6]Department of International Public Health, Liverpool School of Tropical Medicine, Liverpool, UK

**Contributors** All authors contributed to the conceptualisation of the research and contributed to writing the manuscript. M-CVH, MB, JVL, SJ, EHS, DB, CP and JO designed the process evaluation protocol. SJ, MN, SGM and JB led the development of the INTE-AFRICA trial. M-CVH drafted the manuscript and all co-authors edited and commented on subsequent drafts. All authors approved the final draft for submission. All authors agree to be accountable for all aspects of the work in ensuring that questions related to the accuracy or integrity of any part of the work are appropriately investigated and resolved.

**Funding** The INTE-AFRICA project has received funding from the European Union's Horizon 2020 research and innovation programme under grant agreement No 825 698. JVL is supported by a Spanish Ministry of Science, Innovation and Universities Miguel Servet grant (Instituto de Salud Carlos III/ESF, European Union (CP18/00074))

and further acknowledges support to IS Global from the Spanish Ministry of Science, Innovation and Universities through the 'Centro de Excelencia Severo Ochoa 2019–2023' Programme (CEX2018-000806-S), and from the Government of Catalonia through the CERCA Programme.

**Competing interests** None declared.

**Patient and public involvement** Patients and/or the public were involved in the design, or conduct, or reporting, or dissemination plans of this research. Refer to the Methods section for further details.

**Patient consent for publication** Not required.

**Provenance and peer review** Not commissioned; externally peer reviewed.

**ORCID iDs**
Marie-Claire Van Hout http://orcid.org/0000-0002-0018-4060
Max Bachmann http://orcid.org/0000-0003-1770-3506
Jeffrey V Lazarus http://orcid.org/0000-0001-9618-2299

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
