## [Reviewer comments · BMJ Open]

ARTICLE DETAILS

TITLE (PROVISIONAL)	Strengthening integration of chronic care in Africa: protocol for the qualitative process evaluation of integrated HIV, diabetes and hypertension care in a cluster randomised controlled trial in Tanzania and Uganda.
AUTHORS	Van Hout, Marie-Claire; Bachmann, Max; Lazarus, Jeffrey; Shayo, Elizabeth; Bukonya, Dominic; Picchio, Camila; Nyirenda, Moffat; Mfinanga, Sayoki; Birungi, Josephine; Okebe, Joseph; Jaffar, Shabbar

VERSION 1 – REVIEW

REVIEWER	Sonak Pastakia Purdue University (USA and Kenya)
REVIEW RETURNED	03-Jun-2020

GENERAL COMMENTS	The authors present a well written description of the methodology for their qualitative trial describing their evaluation of perspectives from Ugandan and Tanzanian providers on integrating and decentralising care for HIV, diabetes, and hypertension. Their methodology is sound and they have taken great care in being comprehensive in their approach. Specific comments: The introduction is quite long and I think it would be good to shorten it and focus on the main issues at hand for the trial. I think it is well written, just too long Pg 9 line 40, can you also describe the charging dynamics for these different healthcare services? Are all services free? Are all medications free regardless of whether they are hiv medications or htn/dm meds? Who is covering those costs? How does health insurance fit into this as Uganda and TZ have very different dynamics in their health systems. Pg 13 line 17, can you provide further description of the breakdown of the types of healthcare workers you are surveying as I'm sure they have different perspectives based on the cadre. Table three describes this briefly but I'm hoping it covers all of them in sufficient detail. I would note that the supply chain and pharmacy components are not covered despite representing a major piece of the infrastructure required to respond to HIV, dm, and htn in an integrated fashion. I would try to make sure you assess that as well. Can we also assume this only looks at the public sector- I presume it does from the tables and other figures but I think we should state that explicitly. Will you be tapping into the perspective of these providers from their work in the private sector as well? I'm guessing a good
---

	number of the clinicians might have private practices as well. Their responses to your questions might be biased because they might want to protect their private sector interests and have patients with NCD's be referred to them instead of being seen in the public sector where most HIV services are delivered. I would hope your research methodology and focus group questions would assess this dynamic as it is an important consideration which could shade their responses. Might be worthwhile to see if their support for integrated care correlates with whether they practice in the private sector as well. Will participants be compensated? I would assess that as well. I would also like to get a better sense of where the local respondents anticipate the funding for this integration should or could come from. There is a growing body of evidence highlighting that integrated care is the way to go. The main limitation, however, is that funding for this approach is unclear. PEPFAR funding for HIV is potentially going to end within a couple years and become the responsibility of host countries and PEPFAR has consistently shown that they have no appetite to expand the portfolio of services covered by PEPFAR funding. The last thought on the minds of most government officials is expanding the financial burden of care for these patients by including other higher prevalence illnesses. Most countries are trying to plan on how to preserve care for HIV at a minimum. I hope that you can somehow assess these financial aspects especially in your interviews with governmental officials as those are the main barriers to scale up. This is my main concern with the methodology of this trial as the interviews won't actually assess the key issues which are limiting scale up of integrated and decentralised services. I do, however, feel that their methodology should be published.
--	---

REVIEWER	Ozayr Mahomed Discipline of Public Health Medicine 227 George Campbell Building Howard College Campus University of KwaZulu Natal Durban 4051
REVIEW RETURNED	04-Jun-2020

GENERAL COMMENTS	The protocol is scientifically sound. However I am afraid the overelaboration of detail on approach and the continuous repetition of concepts make it difficult to adequately understand the concepts. It maybe more strategic to highlight the aim of the current study after the background. Provide a theoretical framework for the qualitative study without repeating the same in methodology. There is insufficient literature review to substantiate the benefits and sustainability of integrated care as well as implementation of such programs, but there is excess information on motivating why qualitative study design is appropriate and how it will be conducted. The method section need to be modified to provide: Study Setting- accurate information about the facility rather than state one facility- provide current service indicators study design- a single statement highlighting the study design Study population
---

	Data collection methods Data collection variables Analysis Credibility, transferability and viability - The reviewer provided a marked copy with additional comments. Please contact the publisher for full details.
--	---

VERSION 1 – AUTHOR RESPONSE

Reviewer(s)' Comments to Author:

Reviewer: 1

Reviewer Name: Sonak Pastakia

Institution and Country

Purdue University (USA and Kenya)

Please state any competing interests or state 'None declared':

none declared

Please leave your comments for the authors below

The authors present a well written description of the methodology for their qualitative trial describing their evaluation of perspectives from Ugandan and Tanzanian providers on integrating and

decentralising care for HIV, diabetes, and hypertension. Their methodology is sound and they have taken great care in being comprehensive in their approach.

AUTHOR RESPONSE: We wish to thank the reviewer for their very positive and useful comments, all of which are included.

Specific comments:

The introduction is quite long and I think it would be good to shorten it and focus on the main issues at hand for the trial. I think it is well written, just too long

AUTHOR RESPONSE: This section has been shortened with a stronger focus on the sub-Saharan African context.

Pg 9 line 40, can you also describe the charging dynamics for these different healthcare services? Are all services free? Are all medications free regardless of whether they are hiv medications or htn/dm meds? Who is covering those costs? How does health insurance fit into this as Uganda and TZ have very different dynamics in their health systems.

AUTHOR RESPONSE: This information for the two countries has been included.

Pg 13 line 17, can you provide further description of the breakdown of the types of healthcare workers you are surveying as I'm sure they have different perspectives based on the cadre. Table three describes this briefly but I'm hoping it covers all of them in sufficient detail.

AUTHOR RESPONSE: This information has been included in the text and in the Table Three.

I would note that the supply chain and pharmacy components are not covered despite representing a major piece of the infrastructure required to respond to HIV, dm, and htn in an integrated fashion. I would try to make sure you assess that as well. Can we also assume this only looks at the public sector- I presume it does from the tables and other figures but I think we should state that explicitly. Will you be tapping into the perspective of these providers from their work in the private sector as well? I'm guessing a good number of the clinicians might have private practices as well. Their responses to your questions might be biased because they might want to protect their private sector interests and have patients with NCD's be referred to them instead of being seen in the public sector where most HIV services are delivered. I would hope your research methodology and focus group questions would assess this dynamic as it is an important consideration which could shade their

responses. Might be worthwhile to see if their support for integrated care correlates with whether they practice in the private sector as well.

AUTHOR RESPONSE: This information has been included in the text both in terms of the site selection, which are both public facilities, but also on the qualitative investigation. Kindly note that clinical staff work in the public sector, however we will explore this valid point in the interviews with high level policy makers and regional managers.

Will participants be compensated? I would assess that as well.

AUTHOR RESPONSE: This information has been included in the text, regarding compensation for incurred travel to interviews/ focus groups, and provision of refreshments during same.

I would also like to get a better sense of where the local respondents anticipate the funding for this integration should or could come from. There is a growing body of evidence highlighting that integrated care is the way to go. The main limitation, however, is that funding for this approach is unclear. PEPFAR funding for HIV is potentially going to end within a couple years and become the responsibility of host countries and PEPFAR has consistently shown that they have no appetite to expand the portfolio of services covered by PEPFAR funding. The last thought on the minds of most government officials is expanding the financial burden of care for these patients by including other higher prevalence illnesses. Most countries are trying to plan on how to preserve care for HIV at a minimum. I hope that you can somehow assess these financial aspects especially in your interviews with governmental officials as those are the main barriers to scale up. This is my main concern with the methodology of this trial as the interviews won't actually assess the key issues which are limiting scale up of integrated and decentralised services.

AUTHOR RESPONSE: This information has been included in the text, and will be assessed in the qualitative data collection, particularly the interviews with interviews with high level policy makers and regional managers. Health services are already treating diabetes and hypertension, although not as well as HIV which is externally funded. Our hypothesis is that integration will increase efficiency and reduce costs, although better adherence and coverage could increase costs. Interviews with policy makers will include affordability and funding strategy for chronic care.

I do, however, feel that their methodology should be published.

AUTHOR RESPONSE: Thank you.

Reviewer: 2

Reviewer Name

Ozayr Mahomed

Institution and Country

Discipline of Public Health Medicine

227 George Campbell Building

Howard College Campus

University of KwaZulu Natal

Durban

4051

Please state any competing interests or state 'None declared':

Nil

Please leave your comments for the authors below

The protocol is scientifically sound. However I am afraid the overelaboration of detail on approach and the continuous repetition of concepts make it difficult to adequately understand the concepts. It maybe more strategic to highlight the aim of the current study after the background.

AUTHOR RESPONSE: The manuscript has been extensively refocused where required in order to be more concise, and the aim has been positioned after the background on the INTE-AFRICA trial.

Provide a theoretical framework for the qualitative study without repeating the same in methodology.

AUTHOR RESPONSE: The Brofenbrenners ecological model of behaviour is the theoretical model underpinning the process evaluation, and we have made this more explicit by adding in a heading.

There is insufficient literature review to substantiate the benefits and sustainability of integrated care as well as implementation of such programs, but there is excess information on motivating why qualitative study design is appropriate and how it will be conducted.

AUTHOR RESPONSE: We have conducted a detailed scoping review (under submission elsewhere) on the benefits and sustainability of integration in the African context, we have further highlighted this literature, which underpins the need for INTE-AFRICA, and indeed the need for a robust process evaluation to better understand the individual and contextual factors impact on integrated services and the sustainability of such an approach.

The method section need to be modified to provide:

Study Setting- accurate information about the facility rather than state one facility- provide current service indicators.

study design- a single statement highlighting the study design:

Study population:

Data collection methods:

Data collection variables:

Analysis:

Credibility, transferability and viability:

AUTHOR RESPONSE: The methods has been re-organised in the following headings; study setting, study design providing a single statement, followed by detail on the approach itself (see Table Three) and how to works in tandem with collection of selected clinical outcomes (e.g. clinical efficacy of different treatments) and health economic data (e.g. costs and benefits of different approaches) to estimate the

potential benefits to patients and health services at clinic and country level (protocol reported elsewhere); population and recruitment, data collection (the EPP method), data analysis, credibility and transferability. We also provide detail on the two selected facilities.

VERSION 2 – REVIEW

REVIEWER	Ozayr Mahomed Discipline of Public Health Medicine University of KwaZulu Natal Durban South Africa
REVIEW RETURNED	02-Aug-2020

GENERAL COMMENTS	I wish to congratulate the authors on a comprehensive protocol that will provide a template to many budding researchers on the methodology to conduct a qualitative process evaluation. I have one minor comment under study setting: The use of study after cohort may result in confusion between a description of study design and your description of overall study setting In addition to the study limitation of the trial: I suggest that a portion I the protocol speaks directly to the limitation of the study in the form of Selection Bias, Information Bias and Social desirability bias
---

VERSION 2 – AUTHOR RESPONSE

Reviewer(s)' Comments to Author:

Reviewer: 2

Reviewer Name: Ozayr Mahomed

Institution and Country: Discipline of Public Health Medicine, University of KwaZulu Natal, Durban, South Africa

Please state any competing interests or state 'None declared': None declared

Please leave your comments for the authors below

I wish to congratulate the authors on a comprehensive protocol that will provide a template to many budding researchers on the methodology to conduct a qualitative process evaluation.

I have one minor comment under study setting: The use of study after cohort may result in confusion between a description of study design and your description of overall study setting

AUTHOR RESPONSE: We have removed this sentence on page 14. 'We envisage returning to these two clinics in the future, four years and six years after integration, to achieve a deeper understanding of processes and patient and provider experiences'.

In addition to the study limitation of the trial: I suggest that a portion in the protocol speaks directly to the limitation of the study in the form of Selection Bias, Information Bias and Social desirability bias

AUTHOR RESPONSE: We have included the below on page 14/15;

'We recognise the potential for selection and information bias as limitations of the trial itself, and mitigate by using a random sampling approach, defining characteristics in a cohort, using a standardised approach to collecting data with continual assessment of information bias, and ensuring that research personnel are unaware of participant disease status. We will address social desirability in the process evaluation by only providing brief information at the outset of the evaluation in order to avoid priming, using an interview schedule approved by a panel of INTE-AFRICA experts in terms of sensitivity, conducting qualitative research using skilled interviewers with limited power relationship between interviewer and participant, conducting the interviews in a safe and secure setting where the participant feels comfortable, briefing them that there is no right and wrong answer, and finally by encouraging them to use anecdotes and experiential evidence to support their views'.